# Exploiting Verified Neural Networks via Floating Point Numerical Error

## Abstract

Motivated by the need to reliably characterize the robustness of deep neural networks, researchers have developed verification algorithms for deep neural networks. Given a neural network, the verifiers aim to answer whether certain properties are guaranteed with respect to all inputs in a space. However, little attention has been paid to floating point numerical error in neural network verification.

We exploit floating point errors in the inference and verification implementations to construct adversarial examples for neural networks that a verifier claims to be robust with respect to certain inputs. We argue that, to produce sound verification results, any verification system must accurately (or conservatively) model the effects of any float point computations in the network inference or verification system.

## 1 Introduction

Deep neural networks (DNNs) are known to be vulnerable to adversarial inputs (Szegedy et al., 2014), which are images, audio, or texts indistinguishable to human perception that cause a DNN to give substantially different results. This situation has motivated the development of network verification algorithms that claim to prove the robustness of a network (Bunel et al., 2020; Tjeng et al., 2019; Salman et al., 2019), specifically that the network produces identical classifications for all inputs in a perturbation space around a given input.

Verification algorithms typically reason about the behavior of the network assuming real-valued arithmetic. In practice, however, the computation of both the verifier and the neural network is performed on physical computers that use floating point numbers and floating point arithmetic to approximate the underlying real-valued computations. This use of floating point introduces numerical error that can potentially invalidate the guarantees that the verifiers claim to provide. Moreover, the existence of multiple software and hardware systems for DNN inference further complicates the situation, because different implementations exhibit different numerical error characteristics.

We present concrete instances where numerical error leads to unsound verification of real-valued networks. Specifically, we train robust networks on the MNIST and CIFAR10 datasets. We work with the `MIPVerify` complete verifier (Tjeng et al., 2019) and several inference implementations included in the PyTorch (Paszke et al., 2019) framework. For each implementation, we construct image pairs $(x_0, x_{adv})$ where $x_0$ is a brightness modified natural image, such that the implementation classifies $x_{adv}$ differently from $x_0$, $x_{adv}$ falls in a $\ell_\infty$-bounded perturbation space around $x_0$, and the verifier incorrectly claims that no such adversarial image $x_{adv}$ exists for $x_0$ within the perturbation space. Moreover, we show that the incomplete verifier `CROWN` is also vulnerable to floating point error. Our method of constructing adversarial images is not limited to our setting, and it is applicable to other verifiers that do not soundly model floating point arithmetic.

## 2 Background and related work

**Training robust networks:** Researchers have developed various techniques to train robust networks (Madry et al., 2018; Mirman et al., 2018; Tramer & Boneh, 2019; Wong et al., 2020). Madry et al. formulate the robust training problem as minimizing the worst loss within the input perturbation and propose to train robust networks on the data generated by the Projected Gradient Descent

(PGD) adversary (Madry et al., 2018). In this work we consider robust networks trained with the PGD adversary.

**Complete verification:**  The goal of complete verification (a.k.a. exact verification) methods is to either prove the property being verified or provide a counterexample to disprove it. Complete verification approaches have formulated the verification problem as a Satisfiability Modulo Theories (SMT) problem (Scheibler et al., 2015; Huang et al., 2017; Katz et al., 2017; Ehlers, 2017; Bunel et al., 2020) or as a Mixed Integer Linear Programming (MILP) problem (Lomuscio & Maganti, 2017; Cheng et al., 2017; Fischetti & Jo, 2018; Dutta et al., 2018; Tjeng et al., 2019). While SMT solvers are able to model exact floating point arithmetic (Rümmer & Wahl, 2010) or exact real arithmetic (Corzilius et al., 2012), deployed SMT solvers for verifying neural networks all use inexact floating point arithmetic to reason about the neural network inference for efficiency reasons. MILP solvers work directly with floating point, do not attempt to exactly model real arithmetic, and therefore exhibit numerical error. Since floating point arithmetic is not associative, different neural network implementations may produce different results for the same neural network, implying that any sound verifier for this class of networks must reason about the specific floating point error characteristics of the neural network implementation at hand. To the best of our knowledge, no prior work formally recognizes the problem of floating point error in neural network complete verification or exploits floating point error to invalidate verification results.

**Incomplete verification:**  On the spectrum of the tradeoff between completeness and scalability, incomplete methods (a.k.a. certification methods) aspire to deliver more scalable verification by adopting over-approximation, while admitting the inability to either prove or disprove the properties in certain cases. There is a large body of related research (Wong & Kolter, 2017; Weng et al., 2018; Gehr et al., 2018; Zhang et al., 2018; Raghunathan et al., 2018; Dvijotham et al., 2018; Mirman et al., 2018; Singh et al., 2019). Salman et al. (2019) has unified most of the relaxation methods under a common convex relaxation framework. Their results suggest that there is an inherent barrier to tight verification via layer-wise convex relaxation captured by their framework. We highlight that floating point error of implementations that use a direct dot product formulation has been accounted for in some certification frameworks (Singh et al., 2018; 2019) by maintaining upper and lower rounding bounds for sound floating point arithmetic (Miné, 2004). Such frameworks should be extensible to model numerical error in more sophisticated implementations like the Winograd convolution (Lavin & Gray, 2016), but the effectiveness of this extension remains to be studied. Most of the certification algorithms, however, have not considered floating point error and may be vulnerable to attacks that exploit this deficiency.

**Floating point arithmetic:**  Floating point is widely adopted as an approximate representation of real numbers in digital computers. After each calculation, the result is rounded to the nearest representable value, which induces roundoff error. In the field of neural networks, the SMT-based verifier Reluplex (Katz et al., 2017) has been observed to produce false adversarial examples due to floating point error (Wang et al., 2018). The MILP-based verifier MIPVerify (Tjeng et al., 2019) has been observed to give NaN results when verifying pruned neural networks (Guidotti et al., 2020). Such observed floating point unsoundness behavior occurs unexpectedly in running large scale benchmarks. However, no prior work tries to systematically invalidate neural network verification results via exploiting floating point error.

The IEEE-754 (IEEE, 2008) standard defines the semantics of operations and correct rounding behavior. On an IEEE-754 compliant implementation, computing floating point expressions consisting of multiple steps that are equivalent in the real domain may result in different final roundoff error because rounding is performed after each step, which complicates the error analysis. Research on estimating floating point roundoff error and verifying floating point programs has a long history and is actively growing (Boldo & Melquiond, 2017), but we are unaware of any attempt to apply these tools to obtain a sound verifier for any neural network inference implementation. Any such verifier must reason soundly about floating point errors in both the verifier and the neural network inference algorithm. The failure to incorporate floating point error in software systems has caused real-world disasters. For example, in 1992, a Patriot missile missed its target and lead to casualties due to floating point roundoff error related to time calculation (Skeel, 1992).

# 3 PROBLEM DEFINITION

## 3.1 ADVERSARIAL ROBUSTNESS OF NEURAL NETWORKS

We consider 2D image classification problems. Let $\boldsymbol{y} = \mathrm{NN}\left(\boldsymbol{x};\boldsymbol{W}\right)$ denote the classification confidence given by a neural network with weight parameters $W$ for an input $\boldsymbol{x}$, where $\boldsymbol{x} \in \mathbb{R}_{[0,1]}^{m \times n \times c}$ is an image with $m$ rows and $n$ columns of pixels each containing $c$ color channels represented by floating point values in the range $[0, 1]$, and $\boldsymbol{y} \in \mathbb{R}^k$ is a logits vector containing the classification scores for each of the $k$ classes. The class with the highest score is the classification result of the neural network.

For a logits vector $\boldsymbol{y}$ and a target class number $t$, we define the Carlini-Wagner (CW) loss (Carlini & Wagner, 2017) as the score of the target class subtracted by the maximal score of the other classes:

$$L_{\mathrm{CW}}\left(\boldsymbol{y},\,t\right) = y_t - \max_{i \neq t} y_i \qquad (1)$$

Note that $x$ is classified as an instance of class $t$ if and only if $L_{\mathrm{CW}}\left(\mathrm{NN}\left(\boldsymbol{x};\boldsymbol{W}\right),\,t\right) > 0$, assuming no equal scores of two classes.

Adversarial robustness of a neural network is defined for an input $\boldsymbol{x_0}$ and a perturbation bound $\epsilon$, such that the classification result is stable within allowed perturbations:

$$\forall \boldsymbol{x} \in \mathrm{Adv}_\epsilon\left(\boldsymbol{x_0}\right) : L_{\mathrm{CW}}\left(\mathrm{NN}\left(\boldsymbol{x};\boldsymbol{W}\right),\,t_0\right) > 0 \qquad (2)$$
$$\text{where } t_0 = \operatorname{argmax} \mathrm{NN}\left(\boldsymbol{x_0};\boldsymbol{W}\right)$$

In this work we focus on $\ell_\infty$-norm bounded perturbations:

$$\mathrm{Adv}_\epsilon\left(\boldsymbol{x_0}\right) = \{\boldsymbol{x} \mid \|\boldsymbol{x} - \boldsymbol{x_0}\|_\infty \leq \epsilon \,\wedge\, \min \boldsymbol{x} \geq 0 \,\wedge\, \max \boldsymbol{x} \leq 1\} \qquad (3)$$

## 3.2 FINDING ADVERSARIAL EXAMPLES FOR VERIFIED NETWORKS VIA EXPLOITING NUMERICAL ERROR

Due to the inevitable presence of numerical error in both the network inference system and the verifier, the exact specification of $\mathrm{NN}\left(\cdot;\boldsymbol{W}\right)$ (i.e., a bit-level accurate description of the underlying computation) is not clearly defined in (2). We consider the following implementations of convolutional layers included in the PyTorch framework to serve as our candidate definitions of the convolutional layers in $\mathrm{NN}\left(\cdot;\boldsymbol{W}\right)$, and other layers use the default PyTorch implementation:

- $\mathrm{NN_{C,M}}\left(\cdot;\boldsymbol{W}\right)$: A matrix multiplication based implementation on x86/64 CPUs. The convolution kernel is copied into a matrix that describes the dot product to be applied on the flattened input for each output value.
- $\mathrm{NN_{C,C}}\left(\cdot;\boldsymbol{W}\right)$: The default convolution implementation on x86/64 CPUs.
- $\mathrm{NN_{G,M}}\left(\cdot;\boldsymbol{W}\right)$: A matrix multiplication based implementation on NVIDIA GPUs.
- $\mathrm{NN_{G,C}}\left(\cdot;\boldsymbol{W}\right)$: A convolution implementation using the `IMPLICIT_GEMM` algorithm from the cuDNN library (Chetlur et al., 2014) on NVIDIA GPUs.
- $\mathrm{NN_{G,CWG}}\left(\cdot;\boldsymbol{W}\right)$: A convolution implementation using the `WINOGRAD_NONFUSED` algorithm from the cuDNN library (Chetlur et al., 2014) on NVIDIA GPUs. It is based on the Winograd fast convolution algorithm (Lavin & Gray, 2016), which has much higher numerical error compared to others.

For a given implementation $\mathrm{NN_{impl}}\left(\cdot;\boldsymbol{W}\right)$, our method finds pairs of $\left(\boldsymbol{x_0},\,\boldsymbol{x_{adv}}\right)$ represented as single precision floating point numbers such that

1. $\boldsymbol{x_0}$ and $\boldsymbol{x_{adv}}$ are in the dynamic range of images: $\min \boldsymbol{x_0} \geq 0$, $\min \boldsymbol{x_{adv}} \geq 0$, $\max \boldsymbol{x_0} \leq 1$, and $\max \boldsymbol{x_{adv}} \leq 1$.
2. $\boldsymbol{x_{adv}}$ falls in the perturbation space of $\boldsymbol{x_0}$: $\|\boldsymbol{x_{adv}} - \boldsymbol{x_0}\|_\infty \leq \epsilon$
3. The verifier claims that (2) holds for $\boldsymbol{x_0}$
4. $\boldsymbol{x_{adv}}$ is an adversarial image for the implementation: $L_{\mathrm{CW}}\left(\mathrm{NN_{impl}}\left(\boldsymbol{x_{adv}};\boldsymbol{W}\right),\,t_0\right) < 0$

Note that the first two conditions are accurately defined for any implementation compliant with the IEEE-754 standard, because the computation only involves element-wise subtraction and max-reduction that incur no accumulated error. The `Gurobi` (Gurobi Optimization, 2020) solver used by `MIPVerify` operates with double precision internally. Therefore, to ensure that our adversarial examples satisfy the constraints considered by the solver, we also require that the first two conditions hold for $x'_{\text{adv}} = \text{float64} (x_{\text{adv}})$ and $x'_0 = \text{float64} (x_0)$ that are double precision representations of $x_{\text{adv}}$ and $x_0$.

## 3.3 MILP FORMULATION FOR COMPLETE VERIFICATION

We adopt the small CNN architecture from Xiao et al. (2019) and the `MIPVerify` complete verifier of Tjeng et al. (2019) to demonstrate our attack method. We can also deploy our method against other complete verifiers as long as the property being verified involves thresholding continuous variables whose floating point arithmetic is not exactly modeled in the verification process.

The `MIPVerify` verifier formulates the verification problem as an MILP problem for networks composed of linear transformations and piecewise-linear functions (Tjeng et al., 2019). An MILP problem optimizes a linear objective function subject to linear equality and linear inequality constraints over a set of variables, where some variables take real values while others are restricted to be integers. The MILP formulation of the robustness of a neural network involves three parts: introducing free variable $x$ for the adversarial input subject to the constraint $x \in \text{Adv}_\epsilon (x_0)$, formulating the computation $y = \text{NN} (x; W)$, and formulating the attack goal $L_{\text{CW}} (\text{NN} (x; W), t_0) \leq 0$. The network is robust with respect to $x_0$ if the MILP problem is infeasible, and $x$ serves as an adversarial image otherwise. The MILP problem typically optimizes one of the two objective functions: (i) $\min \|x - x_0\|_\infty$ to find an adversarial image closest to $x$, or (ii) $\min L_{\text{CW}} (\text{NN} (x; W), t_0)$ to find an adversarial image that causes the network to produce a different prediction with the highest confidence. Note that although the above constraints and objective functions are nonlinear, most modern MILP solvers can handle them by automatically introducing necessary auxiliary decision variables to convert them into linear forms.

## 4 EXPLOITING A COMPLETE VERIFIER

### 4.1 EMPIRICAL CHARACTERIZATION OF IMPLEMENTATION NUMERICAL ERROR

To guide the design of our attack algorithm we present statistics about numerical error of different implementations.

To investigate end-to-end error behavior, we select an image $x$ and present in Figure 1a a plot of $\|\text{NN} (x + \delta; W) - \text{NN} (x; W)\|_\infty$ against $-10^{-6} \leq \delta \leq 10^{-6}$, where the addition of $x + \delta$ is only applied on the single input element that has the largest gradient magnitude. To minimize the effect of numerical instability due to nonlinearity in the network and focus on fluctuations caused by numerical error, the image $x$ is chosen to be the first MNIST test image on which the network produces a verified robust prediction. We have also checked that the pre-activation values of all the ReLU units do not switch sign. We observe that the change of the logits vector is highly nonlinear with respect to the change of the input, and a small perturbation could result in a large fluctuation. The `WINOGRAD_NONFUSED` algorithm on NVIDIA GPU is much more unstable and its variation is two orders of magnitude larger than the others.

We also evaluate all of the implementations on the whole MNIST test set and compare the outputs of the first layer (i.e., with only one linear transformation applied to the input) against that of $\text{NN}_{\text{C,M}}$, and present the histogram in Figure 1b. It is clear that different implementations usually manifest different error behavior, and again $\text{NN}_{\text{G,CWG}}$ induces much higher numerical error than others.

These observations inspire us to construct adversarial images for each implementation independently by applying small random perturbations on an image close to the robustness decision boundary. We present the details of our method in Section 4.2.

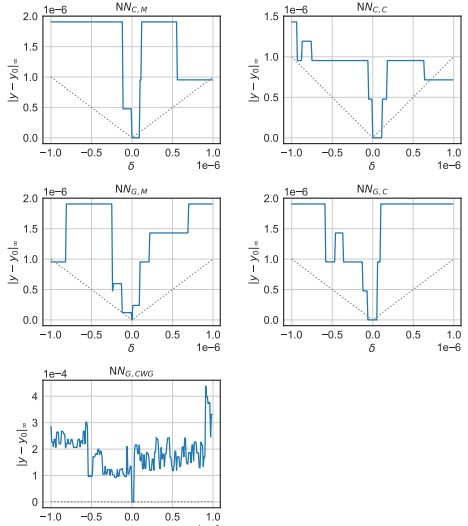

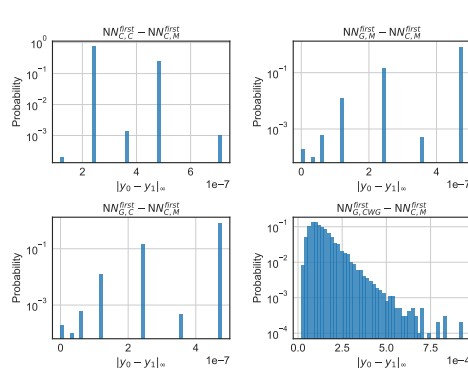

(a) Change of logits vector due to small single-element input perturbations for different implementations. The dashed lines are $y = |\delta|$. This plot shows that the change of output is nonlinear with respect to input changes, and the magnitude of output changes is usually larger than that of input changes. The changes are due to floating point error rather than network nonlinearity because all the pre-activation values of ReLU units do not switch sign.

(b) Distribution of difference relative to $\mathrm{NN_{C,M}}$ of first layer evaluated on MNIST test images. This plot shows that different implementations usually exhibit different floating point error characteristics.

Figure 1: Empirical characterization of numerical error of different implementations

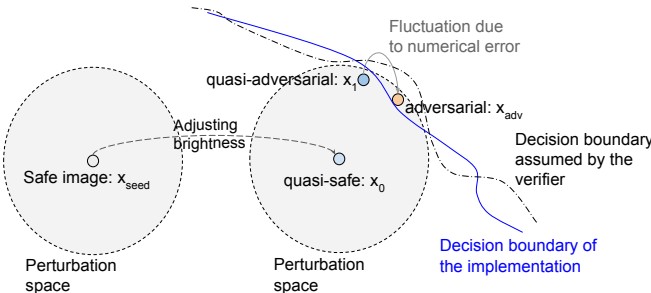

Figure 2: Illustration of our method. Since the verifier does not model the floating point arithmetic details of the implementation, their decision boundaries for the classification problem diverge, which allows us to find adversarial inputs by crossing the boundary via numerical error fluctuations. Note that the verifier usually does not comply with a well defined specification of $\mathrm{NN}\left(\cdot; \boldsymbol{W}\right)$, and therefore it does not define a decision boundary. The dashed boundary in the diagram is just for illustrative purposes.

## 4.2 CONSTRUCTING ADVERSARIAL EXAMPLES

Given a network and weights $\mathrm{NN}\left(\cdot; \boldsymbol{W}\right)$, there exist image pairs $(\boldsymbol{x_0}, \boldsymbol{x_1})$ such that the network is verifiably robust with respect to $\boldsymbol{x_0}$, while $\boldsymbol{x_1} \in \mathrm{Adv}_\epsilon\left(\boldsymbol{x_0}\right)$ and $L_{\mathrm{CW}}\left(\mathrm{NN}\left(\boldsymbol{x_1}; \boldsymbol{W}\right), t_0\right)$ is less than the numerical fluctuation introduced by tiny input perturbations. We call $\boldsymbol{x_0}$ a *quasi-safe image* and $\boldsymbol{x_1}$ the corresponding *quasi-adversarial image*. We then apply small random perturbations on the quasi-adversarial image to obtain an adversarial image. The process is illustrated in Figure 2.

We propose the following proposition for a more formal and detailed description:

**Proposition 1.** *Let $E > 0$ be an arbitrarily small positive number. If a continuous neural network $\mathrm{NN}(\cdot; \boldsymbol{W})$ can produce a verifiably robust classification for class $t$, and it does not constantly classify all inputs as class $t$, then there exists an input $\boldsymbol{x_0}$ such that*

$$0 < \min_{\boldsymbol{x} \in \mathrm{Adv}_\epsilon(\boldsymbol{x_0})} L_{\mathrm{CW}}(\mathrm{NN}(\boldsymbol{x}; \boldsymbol{W}), t) < E$$

*Let $\boldsymbol{x_1} = \mathrm{argmin}_{\boldsymbol{x} \in \mathrm{Adv}_\epsilon(\boldsymbol{x_0})} L_{\mathrm{CW}}(\mathrm{NN}(\boldsymbol{x}; \boldsymbol{W}), t)$ be the minimizer of the above function. We call $\boldsymbol{x_0}$ a quasi-safe image and $\boldsymbol{x_1}$ a quasi-adversarial image.*

*Proof.* Let $f(\boldsymbol{x}) := \min_{\boldsymbol{x'} \in \mathrm{Adv}_\epsilon(\boldsymbol{x})} L_{\mathrm{CW}}(\mathrm{NN}(\boldsymbol{x'}; \boldsymbol{W}), t)$. Since $f(\cdot)$ is composed of continuous functions, $f(\cdot)$ is continuous. Suppose $\mathrm{NN}(\cdot; \boldsymbol{W})$ is verifiably robust with respect to $\boldsymbol{x_+}$ that belongs to class $t$. Let $\boldsymbol{x_-}$ be be any input such that $L_{\mathrm{CW}}(\mathrm{NN}(\boldsymbol{x_-}; \boldsymbol{W}), t) < 0$, which exists because $\mathrm{NN}(\cdot; \boldsymbol{W})$ does not constantly classify all inputs as class $t$. We have $f(\boldsymbol{x_+}) > 0$ and $f(\boldsymbol{x_-}) < 0$, and therefore $\boldsymbol{x_0}$ exists such that $0 < f(\boldsymbol{x_0}) < E$ due to continuity. □

Our method works by choosing $E$ to be a number smaller than the average fluctuation of logits vector introduced by tiny input perturbations as indicated in Figure 1a, and finding a quasi-safe image by adjusting the brightness of a natural image. An adversarial image is then likely to be obtained by applying random perturbations on the corresponding quasi-adversarial image.

Given a particular implementation $\mathrm{NN}_{\mathrm{impl}}(\cdot; \boldsymbol{W})$ and a natural image $\boldsymbol{x_{\mathrm{seed}}}$ which the network robustly classifies as class $t_0$ according to the verifier, we construct an adversarial input pair $(\boldsymbol{x_0}, \boldsymbol{x_{\mathrm{adv}}})$ that meets the constraints described in Section 3.2 in three steps:

1. We search for a coefficient $\alpha \in [0, 1]$ such that $\boldsymbol{x_0} = \alpha \boldsymbol{x_{\mathrm{seed}}}$ serves as the quasi-safe image. Specifically, we require the verifier to claim that the network is robust for $\alpha \boldsymbol{x_{\mathrm{seed}}}$ but not so for $(\alpha - \delta)\boldsymbol{x_{\mathrm{seed}}}$ with $\delta$ being a small positive value. Although the function is not guaranteed to be monotone, we can still use a binary search to find $\alpha$ while minimizing $\delta$ because we only need one such value. However, we observe that in many cases the MILP solver becomes extremely slow for small $\delta$ values, so we start with a binary search and switch to grid search if the solver exceeds a time limit. We set the target of $\delta$ to be $1\mathrm{e}{-7}$ in our experiments and divide the best known $\delta$ to 16 intervals if grid search is needed.

2. We search for the quasi-adversarial image $\boldsymbol{x_1}$ corresponding to $\boldsymbol{x_0}$. We define a loss function with a tolerance of $\tau$ as $L(\boldsymbol{x}, \tau; \boldsymbol{W}, t_0) := L_{\mathrm{CW}}(\mathrm{NN}(\boldsymbol{x}; \boldsymbol{W}), t_0) - \tau$, which can be incorporated in any verifier by modifying the bias of the Softmax layer. We aim to find $\tau_0$ which is the minimal confidence of all images in the perturbation space of $\boldsymbol{x_0}$, and $\tau_1$ which is slightly larger than $\tau_0$ with $\boldsymbol{x_1}$ being the corresponding adversarial image:

$$\begin{cases} \forall \boldsymbol{x} \in \mathrm{Adv}_\epsilon(\boldsymbol{x_0}): \; L(\boldsymbol{x_0}, \tau_0; \boldsymbol{W}, t_0) > 0 \\ \boldsymbol{x_1} \in \mathrm{Adv}_\epsilon(\boldsymbol{x_0}) \\ L(\boldsymbol{x_1}, \tau_1; \boldsymbol{W}, t_0) < 0 \\ \tau_1 - \tau_0 < 1\mathrm{e}{-7} \end{cases}$$

Note that $\boldsymbol{x_1}$ is produced by the complete verifier as a proof for nonrobustness given the tolerance $\tau_1$. The above values are found via a binary search with initialization $\tau_0 \leftarrow 0$ and $\tau_1 \leftarrow \tau_{\mathrm{max}}$ where $\tau_{\mathrm{max}} := L_{\mathrm{CW}}(\mathrm{NN}(\boldsymbol{x_0}; \boldsymbol{W}), t_0)$. If the verifier is able to compute the *worst* objective $\tau_w = \min_{\boldsymbol{x} \in \mathrm{Adv}_\epsilon(\boldsymbol{x_0})} L_{\mathrm{CW}}(\mathrm{NN}(\boldsymbol{x}; \boldsymbol{W}), t_0)$, the binary search can be accelerated by initializing $\tau_0 \leftarrow \tau_w - \delta_s$ and $\tau_1 \leftarrow \tau_w + \delta_s$. We empirically set $\delta_s = 3\mathrm{e}{-6}$ to incorporate the numerical error in the verifier so that $L(\boldsymbol{x_0}, \tau_w - \delta_s; \boldsymbol{W}, t_0) > 0$ and $L(\boldsymbol{x_0}, \tau_w + \delta_s; \boldsymbol{W}, t_0) < 0$. The binary search is aborted if the solver times out.

3. We minimize $L_{\mathrm{CW}}(\mathrm{NN}(\boldsymbol{x_1}; \boldsymbol{W}), t_0)$ with hill climbing via applying small random perturbations on the quasi-adversarial image $\boldsymbol{x_1}$ while projecting back to $\mathrm{Adv}_\epsilon(\boldsymbol{x_0})$ to find an adversarial example. The perturbations are applied on patches of $\boldsymbol{x_1}$, as described in Appendix A. The random perturbations are on the scale of $2\mathrm{e}{-7}$, corresponding to the input perturbations that cause a change in Figure 1a.

### 4.3 EXPERIMENTS

We conduct our experiments on a workstation equipped with two GPUs (NVIDIA Titan RTX and NVIDIA GeForce RTX 2070 SUPER), 128 GiB of RAM and an AMD Ryzen Threadripper 2970WX

Table 1: Number of successful adversarial attacks for different neural network implementations. The number of quasi-adversarial images in the first column corresponds to the cases where the solver does not time out at the initialization step. For each implementation, we try to find adversarial images by applying random perturbations on each quasi-adversarial image and report the number of successfully found adversarial images here.

| | #quasi-adv / #tested | $NN_{C,M}$ | $NN_{C,C}$ | $NN_{G,M}$ | $NN_{G,C}$ | $NN_{G,CWG}$ |
|---|---|---|---|---|---|---|
| MNIST | 18 / 32 | 2 | 3 | 1 | 3 | 7 |
| CIFAR10 | 26 / 32 | 16 | 12 | 7 | 6 | 25 |

verified robust 7 $L_{CW}=2.5$ | $NN_{C,M}$ 2 $L_{CW}=-3.6e-07$ | $NN_{C,C}$ 2 $L_{CW}=-3.6e-07$ | $NN_{G,M}$ 2 $L_{CW}=-1.2e-07$ | $NN_{G,C}$ 2 $L_{CW}=-2.4e-07$ | $NN_{G,CWG}$ 2 $L_{CW}=-6.9e-04$

airplane $L_{CW}=0.5$ | horse $L_{CW}=-1.8e-06$ | horse $L_{CW}=-3.5e-06$ | horse $L_{CW}=-1.5e-06$ | horse $L_{CW}=-3.5e-06$ | horse $L_{CW}=-7.0e-04$

horse $L_{CW}=0.5$ | deer $L_{CW}=-1.2e-06$ | deer $L_{CW}=-1.5e-06$ | deer $L_{CW}=-1.2e-07$ | deer $L_{CW}=-1.2e-07$ | deer $L_{CW}=-2.4e-04$

Figure 3: The quasi-safe images with respect to which all implementations are successfully attacked, and corresponding adversarial images

24-core processor. We train the small architecture from Xiao et al. (2019) with the PGD adversary and the RS Loss on MNIST and CIFAR10 datasets. The trained networks achieve 94.63% and 44.73% provable robustness with perturbations of $\ell_\infty$ norm bounded by 0.1 and 2/255 on the two datasets respectively, similar to the results reported in Xiao et al. (2019). Our code will be made publicly available after the review process.

Although our method only needs $O(-\log \epsilon)$ invocations of the verifier where $\epsilon$ is the gap in the binary search, the verifier is too slow to run a large benchmark in a reasonable time. Therefore, for each dataset we only test our method on 32 images randomly sampled from the verifiably robustly classified test images. The time limit of MILP solving is 360 seconds. Out of these 32 images, we have successfully found quasi-adversarial images ($x_1$ from Section 4.2 Step 2, where failed cases are solver timeouts) for 18 images on MNIST and 26 images on CIFAR10. We apply random perturbations to these quasi-adversarial images to obtain adversarial images within the perturbation range of the quasi-safe image ($x_0 = \alpha x_{seed}$ from Section 4.2 Step 1). All the implementations that we have considered are successfully attacked. We present the detailed numbers in Table 1. We also present in Figure 3 the quasi-safe images on which our attack method succeeds for all implementations and the corresponding adversarial images.

# 5 EXPLOITING AN INCOMPLETE VERIFIER

The relaxation adopted in certification methods renders them incomplete but also makes their verification claims more robust to floating point error compared to complete verifiers. In particular, we evaluate the CROWN framework (Zhang et al., 2018) on our randomly selected test images and

corresponding quasi-safe images from Section 4.3. CROWN is able to verify the robustness of the network on 29 out of the 32 original test images, but it is unable to prove the robustness for any of the quasi-safe images. Note that MIPVerify claims that the network is robust with respect to all the original test images and corresponding quasi-safe images.

Given the above situation, we demonstrate that incomplete verifiers are still prone to floating point error. We build a neural network that takes a $13 \times 13$ single-channel input image, followed by a $5 \times 5$ convolutional layer with a single output channel, two fully connected layers with 16 output neurons each, a fully connected layer with one output neuron denoted as $u = \max(\boldsymbol{W_u}\boldsymbol{h_u} + b_u, 0)$, and a final linear layer that computes $\boldsymbol{y} = [u, 1e-7]$ as the logits vector. All the hidden layers have ReLU activation. The input $\boldsymbol{x_0}$ is taken from a Gaussian distribution. The hidden layers have random Gaussian coefficients, and the biases are chosen so that (i) the ReLU neurons before $u$ are always activated for inputs in the perturbation space of $\boldsymbol{x_0}$, (ii) $u = 0$ always holds for these inputs, and (iii) $b_u$ is maximized with all other parameters fixed. CROWN is able to prove that all ReLU neurons before $u$ are always activated but $u$ is never activated, and therefore it claims that the network is robust with respect to perturbations around $\boldsymbol{x_0}$. However, by initializing the quasi-adversarial input $\boldsymbol{x_1} \leftarrow \boldsymbol{x_0} + \epsilon \operatorname{sign}(\boldsymbol{W_{equiv}})$ where $\boldsymbol{W_{equiv}}$ is the product of all the coefficient matrices of the layers up to $u$, we successfully find adversarial inputs for all the five implementations considered in this work by randomly perturbing $\boldsymbol{x_1}$ in a way similar to Step 3 of Section 4.2.

## 6 DISCUSSION

We agree with the security expert Window Snyder, "One single vulnerability is all an attacker needs". Unfortunately, most previous work on neural network verification abstains from discussing possible vulnerabilities in their methods. We have demonstrated that neural network verifiers, although meant to provide *security guarantees*, are systematically exploitable. The underlying tradeoff between soundness and scalability in the verification of floating point programs is fundamental but has not received enough attention in the neural network verification literature.

One appealing remedy is to introduce floating point error relaxations into complete verifiers, such as by verifying for a larger $\epsilon$ or setting a threshold for accepted confidence score. However, a tight and sound relaxation is extremely challenging to find. We are unaware of prior attempt to formally prove error bounds for practical and accelerated neural network implementations or verifiers.

Some incomplete verifiers have incorporated floating point error by maintaining upper and lower rounding bounds of internal computations (Singh et al., 2018; 2019), which is also potentially applicable to complete verifiers. However, this approach relies on the specific implementation details of the inference algorithm — optimizations such as Winograd (Lavin & Gray, 2016) or FFT (Abtahi et al., 2018) would either invalidate the robustness guarantees or require changes to the analysis algorithm.

Another approach is to quantize the computation to align the inference implementation with the verifier. For example, if we require all activations to be multiples of $s_0$ and all weights to be multiples of $s_1$, where $s_0 s_1 > 2E$ and $E$ is a very loose bound of possible implementation error, then the output can be rounded to multiples of $s_0 s_1$ to completely eliminate numerical error. Binarized neural networks (Hubara et al., 2016) are a family of extremely quantized networks, and their verification (Narodytska et al., 2018; Shih et al., 2019) is sound and complete. However, the problem of robust training and verification of quantized neural networks (Jia & Rinard, 2020) is relatively under-examined compared to that of real-valued neural networks (Madry et al., 2018; Mirman et al., 2018; Tjeng et al., 2019; Xiao et al., 2019).

## 7 CONCLUSION

Floating point error should not be overlooked in the verification of real-valued neural networks, as we have presented techniques that construct adversarial examples for neural networks claimed to be robust by a verifier. We hope our results will help to guide future neural network verification research by providing another perspective for the tradeoff between soundness, completeness, and scalability.

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

## A    RANDOM PERTURBATION ALGORITHM

We present the details of our random perturbation algorithm below. Note that the Winograd convolution computes a whole output patch in one iteration, and therefore we handle it separately in the algorithm.

---

**Input:** quasi-safe image $x_0$
**Input:** target class number $t$
**Input:** quasi-adversarial image $x_1$
**Input:** input perturbation bound $\epsilon$
**Input:** a neural network inference implementation $\text{NN}_{\text{impl}}(\cdot; W)$
**Input:** number of iterations $N$ (default value 1000)
**Input:** perturbation scale $u$ (default value 2e$-$7)
**Output:** an adversarial image $x_{\text{adv}}$ or FAILED

**for** Index $i$ of $x_0$ **do**             ▷ Find the weakest bounds $x_l$ and $x_u$ for allowed perturbations
    $x_l[i] \leftarrow \max(\text{nextafter}(x_0[i] - \epsilon, 0), 0)$
    $x_u[i] \leftarrow \min(\text{nextafter}(x_0[i] + \epsilon, 1), 1)$
    **while** $x_0[i] - x_l[i] > \epsilon$ **or** $\text{float64}(x_0[i]) - \text{float64}(x_l[i]) > \epsilon$ **do**
        $x_l[i] \leftarrow \text{nextafter}(x_l[i], 1)$
    **end while**
    **while** $x_u[i] - x_0[i] > \epsilon$ **or** $\text{float64}(x_u[i]) - \text{float64}(x_0[i]) > \epsilon$ **do**
        $x_u[i] \leftarrow \text{nextafter}(x_u[i], 0)$
    **end while**
**end for**

**if** $\text{NN}_{\text{impl}}(\cdot; W)$ is $\text{NN}_{\text{G,CWG}}(\cdot; W)$ **then**
    (offset, stride) $\leftarrow (4, 9)$          ▷ The Winograd algorithm in cuDNN produces $9 \times 9$ output tiles for $13 \times 13$ input tiles and $5 \times 5$ kernels. The offset and stride here ensure that perturbed tiles contribute independently to the output.
**else**
    (offset, stride) $\leftarrow (0, 4)$          ▷ Work on small tiles to avoid random errors get cancelled
**end if**

**for** $i \leftarrow 1$ **to** $N$ **do**
    **for** $(h, w) \leftarrow (0, 0)$ **to** $(\text{height}(x_1), \text{width}(x_1))$ **step** $(\text{stride, stride})$ **do**
        $\delta \leftarrow \text{uniform}(-u, u, (\text{stride} - \text{offset}, \text{stride} - \text{offset}))$
        $x_1' \leftarrow x_1[:]$
        $x_1'[h + \text{offset} : h + \text{stride}, w + \text{offset} : w + \text{stride}] += \delta$
        $x_1' \leftarrow \max(\min(x_1', x_u), x_l)$
        **if** $L_{\text{CW}}(\text{NN}_{\text{impl}}(x_1'; W), t) < L_{\text{CW}}(\text{NN}_{\text{impl}}(x_1; W), t)$ **then**
            $x_1 \leftarrow x_1'$
        **end if**
    **end for**
**end for**
**if** $L_{\text{CW}}(\text{NN}_{\text{impl}}(x_1; W), t) < 0$ **then**
    **return** $x_{\text{adv}} \leftarrow x_1$
**else**
    **return** FAILED
**end if**

---

