# OpenReview forum: "Exploiting Verified Neural Networks via Floating Point Numerical Error"
_ICLR.cc/2021/Conference — Reject_

### Official Review · AnonReviewer2 · 2020-10-25
**Verification of neural networks for computer vision tasks is not well defined. Hence, analyzing attack models on the verification engines is not well grounded.**

**Rating:** 3
**Confidence:** 5

**Review:**

In the recent literature there has been a rise in the number of papers which attempt to verify neural networks. The specification of the verification problems often gets adapted according to the application in mind. More specifically, for image classification networks, the problem is to prove that the output of the neural network does not flip for small perturbations to the pixel values. For a robotic setting, the problem is often safety and convergence to some goal state. Where the neural network operates in closed loop with the system dynamics.

The authors in this paper present an adversarial attack model on neural networks, which is deemed correct by some verifier. More specifically , given a neural network which can be shown to be robust to adversarial perturbations around some input, the authors exploit numerical errors in the computations to attack the network. Demonstrating the presence of loop holes in the proving engines itself. This is due to the approximation errors introduced by using floating point numbers.

In my opinion, the notion of input sets in the space of images, is not a very useful one. Mainly because the interval valued sets representing  perturbations of the input image, is far removed from the intended specification.  It's a step in the right direction, if the verification of computer vision task was a well defined problem. Since it's not clear what to verify in the first place, the use case of this paper  is not a very convincing one in my opinion. The problem of verifying neural networks in a robotic setting has a more meaningful specification.  Hence, i don't think that this paper in itself will be interesting to the general theme of the conference.

---

> ### Author Response · Authors · 2020-11-13
> **Verifying l-inf robustness for computer vision tasks is widely accepted**
>
> The only criticism appears to be related to the choice of the correctness specification (specifically, the robustness against $\ell_\inf$-bounded perturbations). The reviewer appears to reject this specification as not useful. There is no other criticism of the research in this review.
>
> This robustness specification and similar ones are now widely accepted and used within the field, and related research using these specifications has produced interesting and useful results. For example, many existing NN verifiers,  including both complete and incomplete ones, such as MIPVerify, ERAN, and CROWN, have targeted the robustness of computer vision NNs against $\ell_\inf$-bounded perturbations as an application. As another example, Madry et al. (2018) demonstrate their PGD robust training method using this $\ell_\inf$ robustness formulation. Their following work (*Adversarial Examples Are Not Bugs, They Are Features*) that analyzes robustness with a similar specification (based on $\ell_2$ distance) proposes robust features in datasets, which are better aligned with human perception.
>
> While we agree that such specifications may not fully capture every desirable robustness criteria, they provide a baseline that, if violated, highlights the vulnerability of even verified networks to adversarial attacks. Moreover, our work is not specifically tailored for this specification. Our most important message, that verifiers should soundly model FP computations, also applies to the verification of other specifications in other settings.
>
> Many papers with similar settings have been published in ICLR, such as [this one](https://openreview.net/forum?id=rJzIBfZAb),  [that one](https://openreview.net/forum?id=HyGIdiRqtm) and [another one](https://openreview.net/forum?id=BJfIVjAcKm). This fact suggests that this setting, which is the robustness of deep neural networks for computer vision tasks under $\ell_\inf$-bounded perturbations, is relevant to the general theme of the conference. .

---

### Official Review · AnonReviewer3 · 2020-10-28
**A service to the community**

**Rating:** 8
**Confidence:** 5

**Review:**

The paper presents a method to find adversarial inputs for neural networks in regions where the networks can be "proven" not to admit any such adversarial examples, practically demonstrating the unsoundness of a "complete verifier" as well as an "incomplete verifier".
While it was already obvious to me that "verifiers" that assume floating-point arithmetic is the same as real arithmetic are unsound, the paper is a service to the community in that it also makes this very obvious to informed outsiders who may not have already questioned the validity of robustness verification research that does not model round-off and even ignores it in its own implementation. The related work section does a good job of surveying the state of the art as it relates to floating-point soundness. The authors also took some space to discuss how their findings relate to current and future research on robustness verification, which I think is important in this case.

Perhaps there could be a short discussion of challenges that different approaches face to become sound with respect to floating-point semantics. (For example, it seems particularly challenging for approaches based on duality, as the correctness of certificates depends non-trivially on closed-form solutions to optimization problems as well as associativity of addition.)

The technical sections are mostly well-written, though I was not able to figure out some details. For example, it is not so clear how precisely binary search is used to find α and δ simultaneously. Section 4.2 is a bit dense and its presentation could probably be improved.


"inevitable presence of numerical error in [...] the verifier".
It is not inevitable that the verifier is subject to "error". We could encode the precise floating-point semantics of the neural network as a SAT formula (and then watch the SAT solver time out, but this does give a sound and complete method).

---

> ### Author Response · Authors · 2020-11-13
> **Very helpful comments**
>
> Thanks so much for those helpful comments!
>
> > There could be a short discussion of challenges that different approaches face to become sound with respect to floating-point semantics.
>
> Thanks for providing this suggestion. We will include related discussions in our next revision.
>
> > For example, it is not so clear how precisely binary search is used to find $\alpha$ and $\delta$ simultaneously. Section 4.2 is a bit dense and its presentation could probably be improved.
>
> We maintain the lower bound $\alpha_0$ and the upper bound $\alpha_1$ in binary search, where $\delta = \alpha_1 - \alpha_0$. Ideally we'd like to repeat the binary search until $\delta$ decreases to a predefined threshold, but it does not always work due to solver timing out. Therefore, we only conduct the binary search to minimize $\delta$ as much as possible, and switch to grid search if the solver times out.
>
> Thanks for pointing out this confusing part. We will improve the algorithm presentation in the next revision.
>
> >  It is not inevitable that the verifier is subject to "error".
>
> That's right! We have also discussed the possibility of using exact floating point or real arithmetic in SMT solvers. In the context of this paragraph, we are referring to "practical" solvers that scale nontrivially, which can not afford the complexity of exact FP modeling.
>
> Thanks for pointing out this inaccurate sentence. We'll improve it in the next revision.

---

### Official Review · AnonReviewer1 · 2020-10-28
**Good presentation but problem of limited impact.**

**Rating:** 4
**Confidence:** 5

**Review:**

Summary:
The authors develop a method to generate pairs of sample that are separated by a small adversarial perturbation, that have different class, but with the specificity that the a complete verifier would returns a result indicating that this sample admits no adversarial perturbation (despite the fact that it does, as evidenced by the second element of the pair).
These samples are obtained by considering a brightness perturbation of the image and finding the parameter (alpha) at which the verifier switch from returning "safe" to "unsafe". The resulting perturbed image is going to have adversarial examples very close to the boundary of the region considered, so small floating point errors might result in returning incorrect results.

Main thoughts:
The problem that the author discuss is very well highlighted and explained. It is clear what vulnerability they identified, as well as the mechanism that they use to highlight it.
On the other hand, in terms of importance, I would rank it more as an interesting observation that an actual critical problem. If we assume, that what I'm caring is robustness of my image classification system for perturbation of size epsilon=0.1, then it seems that the worst that can happen is that some samples that I verified to be robust for epsilon=0.1, are in practice only robust for epsilon=0.09999? This doesn't seem overtly critical and would result in essentially the same result in any application.

Questions:
- The choice of what solver to use as a backend for a MIP formulation of the Neural Network verification problem is an implementation detail. MIPVerify could well be implemented with a different solver? (MIP solvers returning incorrect result due to floating point errors is not a new problem and there seems to be some literature in how to adress these problems if they are considered of importance "Safe bounds in linear and mixed-integer linear programming, Neumaier & Shcherbina")
In addition, could this problem be solved by simply adjusting the tolerance parameters of the solver? I did not see any discussion of this by the authors, but I imagine that the default parameters used by the verifier might be geared more towards speed than towards perfect accuracy.
- The authors mention verifiers that incorporate proper handling of floating point errors (ERAN) but then reject it by saying that it rely on specific implementation details of the inference algorithm. This seems strange because that's exactly the recommendation that the authors make. Page 2: "any sound verifier for this class of networks must reason about the specific floating point error characteristics of the neural network implementation at hand."

Minor questions:
In Figure 1.a, it seems like for the first 4 graphs, the dotted lines which I assume implies what the difference should be are lines with slope 1. Why would the change in the logit vector vary at the same rate as the perturbations? Shouldn't there be a slope dependent on the corresponding gradient coefficient?

---

> ### Author Response · Authors · 2020-11-13
> **The impact is that some verfiers are shown to be untrustworthy**
>
> Thanks so much for the constructive criticism!
>
> ## Regarding the technical concerns
>
> > MIPVerify could well be implemented with a different solver?
>
> It might be possible to compute the pre-relu bounds conservatively in MIPVerify and also use a safe solver such as the one discussed in [1]. However, there are potentially nontrivial barriers regarding adopting such methods:
>
> 1. Directed rounding to maintain FP soundness poses implementation and engineering challenges. In fact, [1] presents only a theoretical analysis and does not conduct experiments to evaluate their methods. The loss of scalability needs to be investigated.
> 2. The directed rounding only works with a straightforward NN implementation that computes the dot product sequentially. Further research is needed to address how to adapt the method to verify other implementations. A possible option is to completely formulate the implementation computation in the MILP formulation, but that might be very slow.
> 3. The conservative formulation renders the verification incomplete. The exact loss of completeness needs to be evaluated.
>
>
> We are happy to include a brief discussion in the next version.
>
> [1] Neumaier, Arnold, and Oleg Shcherbina. "Safe bounds in linear and mixed-integer linear programming." Mathematical Programming 99.2 (2004): 283-296.
>
> > Could this problem be solved by simply adjusting the tolerance parameters of the solver?
>
> It's challenging to obtain a sound tolerance. The error due to FP unsoundness can be scaled by malicious networks (for example, multiplying $y$ by $1e7$ in Section 5).
>
> > ERAN is rejected by saying that it rely on specific implementation details of the inference algorithm
>
> ERAN is sound with respect to the kind of straightforward NN implementation considered in their formulation. It can be unsound for other accelerated implementations (such as Winograd) that are widely adopted. How to adapt it to other implementations has not been discussed by the original authors and calls for further research. Here we are not rejecting ERAN, but emphasizing that a verifier should explicitly specify its targeted implementation(s), and being versatile for many implementations is an appealing feature.
>
> We will clarify in the next version.
>
> > Why would the change in the logit vector vary at the same rate as the perturbations? Shouldn't there be a slope dependent on the corresponding gradient coefficient?
>
> The dotted line is a reference for the "identity change" and can be helpful in emphasizing that output change is random with respect to input change and showing that they are of the same magnitude in most cases. It is not intended to indicate what the difference should be. The change is random due to FP error and does not depend on the gradient.
>
> We will clarify in the next version.
>
> ## Regarding the impact
>
> > It seems that the worst that can happen is that some samples that I verified to be robust for epsilon=0.1, are in practice only robust for epsilon=0.09999?
>
> It can not be proved that the gap of 0.00001 is sound for the target implementation and the verifier.
>
> The most important takeaway from our paper is that verifiers that ignore FP error provide **no correctness guarantees**. While we work on naturally trained networks for which the gap between the verifiable perturbation bound and the true robust perturbation bound might be small, in practice it is totally possible for an adversary to manipulate network weights and architecture to make this gap significantly larger.
>
> We hope our work raises the awareness regarding FP soundness among researchers, and future NN verifiers can incorporate FP soundness into their design.

---

> > ### Comment · AnonReviewer1 · 2020-11-16
> > **Thank you for the replies.**
> >
> > I thank the reviewers for their detailed answers to my questions.
> >
> > I understand the interest of the paper from a purist point of view (Complete verifiers should not make mistakes) but I still remain unconvinced of the practical importance. Adversarial examples were shocking because we discovered that tiny perturbations were leading to extremely bad predictions so it could be imagined what the bad consequences would be.
> > In the case presented here, I don't think that a strong case is made for potential problems. Unless I'm mistaken (and I would welcome the authors showing me that I'm wrong), what is found are "samples that are at the edge of being robust" and that due to floating point inaccuracies, errors might happen but there is in practice no strong difference between a sample robust for 0.1 and a sample robust for 0.099999.
> >
> > Regarding the point that "in practice, it is totally possible for an adversary to manipulate network weights and architectures to make this gap significantly larger":
> > - At the point where the adversary manipulates the weights, the architecture, and the inputs, I'm a bit confused as to what can you reasonably expect to do.
> >
> > All in all, my thinking remains mostly the same: the phenomenon described in this paper is an interesting curiosity, and the authors describe it well, but the practical importance is actually limited, especially given that for people actually caring about floating point soundness, solutions like ERAN actually exist even if they come with certain trade-offs.

---

> > > ### Author Response · Authors · 2020-11-16
> > > **Thanks for clearly stating the concerns**
> > >
> > > Thanks for replying so quickly and clearly stating your concerns regarding the impact.
> > >
> > > > At the point where the adversary manipulates the weights, the architecture, and the inputs, I'm a bit confused as to what can you reasonably expect to do
> > >
> > > What an adversary can do depends on the threat model. The paper demonstrates a threat model where the adversary does not have control of the weights and the architecture. The paper shows that even this weak adversary can exploit a verified network. A stronger adversary will find it even easier to exploit the network (as we have shown in Section 5 for incomplete verifiers). In particular, we believe that there are realistic scenarios in which an adversary can manipulate the weights, specifically scenarios in which a system accepts untrusted neural networks that are verified to satisfy certain properties. Because neural networks are so opaque, it can be difficult to detect the presence of such manipulation.
> > >
> > > Once again, there are no guarantees whatsoever provided by any exact floating point verifier currently extant. We have demonstrated that existing floating point verifiers can be exploited. It is not the case that anyone can *trust* any floating point verification because there are no guarantees. An adversary can exploit this fact in multiple ways depending on how such a system is actually deployed.
> > >
> > > > What is found are "samples that are at the edge of being robust"
> > >
> > > We'd like to summarize the result as "samples at the edge of being robust can be effectively synthesized to change the network decision, which breaks the security guarantees provided by verifiers in practice". We can imagine that such samples exist, but showing that they can be easily found makes a practical impact.
> > >
> > > > Due to floating point inaccuracies, errors might happen but there is in practice no strong difference between a sample robust for 0.1 and a sample robust for 0.099999.
> > >
> > > We emphasize once again that there are *no guarantees* provided by any floating point complete verifier currently extant. There is no guarantee at all that one can trust a network claimed to be robust at one epsilon is actually robust at a somewhat smaller epsilon. This is particularly problematic for safety-critical scenarios or scenarios that are making consequential decisions about people’s lives.
> > >
> > > Moreover, as we have argued in previous discussions, the difference might be much larger than 0.1 vs 0.099999 if the network has certain properties (for example, as stated in the paper, MIPVerify has been observed to give NaN results when verifying pruned neural networks (Guidotti et al., 2020)) or if the system is deployed in certain ways that allow malicious network weights to be uploaded.
> > >
> > > > Especially given that for people actually caring about floating point soundness, solutions like ERAN actually exist even if they come with certain trade-offs.
> > >
> > > These tradeoffs are important and consequential. Their use of sound over-approximation may lead to overly conservative results that are less useful in practice. And as discussed further below, they target only a specific NN implementation (the straightforward dot-product implementation) and do not verify more efficient implementations that are widely used in practice.
> > >
> > > More importantly, if someone is not consciously aware of FP soundness issues in verifiers, they may get a false sense of safety from verifiers from which they are expecting an absolute guarantee. Unfortunately, the acknowledgement of FP soundness issues seems quite limited, especially among ML researchers, as can be seen from the following facts:
> > >
> > > 1. Many verifiers, such as MIPVerify, Reluplex, and CROWN, do not discuss the potential impact of their FP unsoundness. The reviews of MIPVerify [are accessible](https://openreview.net/forum?id=HyGIdiRqtm), and none of the reviewers raised any concerns regarding FP soundness.
> > > 2. Few verifiers devote any effort to being sound. ERAN has done an excellent job. ReluVal also makes a contribution by rounding the interval estimations outward, but it does not seem to take care of internal computations and uses OpenBLAS for fast matrix multiplication (which produces approximate results without guaranteed error bound). We are unaware of others besides these two.
> > > 3. Even ERAN does not discuss the possibility of sound verification for other NN implementations. A user currently has no choice if they want to obtain sound verification for accelerated implementations such as Winograd (or worse, the user may be unaware of the unsoundness of ERAN for such implementations due to the lack of related discussions).

---

### Official Review · AnonReviewer4 · 2020-10-29
**Interesting Paper but lacks sufficient contribution**

**Rating:** 4
**Confidence:** 3

**Review:**

This paper focuses on the floating point unsoundness of neural network verification procedures, and shows that the results from such verifiers can not be trusted. To drive home the message of the paper, the authors take MIPVerify (which doesn't ensure FP soundness) and shows that they are able to construct adversarial examples for the cases that are returned as verified by MIPVerify.

It is an interesting nice paper but the contribution is weakened by two  facts: One, it is already known that MIPVerify doesn't ensure FP soundness; which is also acknowledged by its authors. Second, when FP soundness is not ensured, and given the fact that adversarial examples are widely present, it is no surprise that one could find adversarial examples.

So I am split on the paper. From a formal methods perspective, the discovery of the paper is not surprising as Floating point computations are known to be important (this is one of the reasons that SMT solvers put a lot of emphasis on FP). But perhaps from ML practitioner, it may be interesting.

---

> ### Author Response · Authors · 2020-11-13
> **We are the first to implement such an attack**
>
> Thanks for your comments.
>
> > One, it is already known that MIPVerify doesn't ensure FP soundness; which is also acknowledged by its authors.
>
> The FP soundness problem has not been systematically addressed in previous work. Previous work has not gone further than reporting the accidentally found unsound results, while we discuss the fundamental tradeoff between soundness and scalability, and show that FP unsoundness can be systematically exploited to invalidate verification claims.
>
> > Second, when FP soundness is not ensured, and given the fact that adversarial examples are widely present, it is no surprise that one could find adversarial examples.
>
> We are the first to actually present a practical method that can systematically construct adversarial examples for networks claimed to be robust by a complete verifier via exploiting FP error, which we believe is an important discovery that further highlights the unsoundness of the verifiers for FP neural networks. As a not-so-proper analogy of this situation, it is unsurprising to acknowledge that the factorization of big integers exists, but presenting a practical factorization method invalidates the security guarantees of many cryptography algorithms.

---

> > ### Comment · AnonReviewer4 · 2020-11-16
> > **The significance of contribution?**
> >
> > I apologize for coming across as harsh but I truly fail to see value in finding an adversarial example in the context of neural networks when the given technique certifies "Sound" under the explicit acknowledgment that such a certificate does not assume FP soundness. The fact that a program may be certified sound under the assumption of reals may indeed be unsound is folklore in verification (hence the push for FP soundness in the first place). It would have indeed been a real surprise if somehow neural networks don't exhibit such a widely observed phenomenon i.e., if the paper could show that in the case of neural networks, FP unsound techniques suffice.
> >
> > I will leave to AC to ensure consistency with respect to another concurrent submission.

---

> > > ### Author Response · Authors · 2020-11-17
> > > **An inaccurate portrait of the situation**
> > >
> > > Thanks for clearly expressing your concerns. However, we still believe that our paper is significant in the context of NN verification research.
> > >
> > > > The fact that a program may be certified sound under the assumption of reals may indeed be unsound is folklore in verification (hence the push for FP soundness in the first place).
> > >
> > > Unfortunately, this is an inaccurate portrait of the current reality, especially for the NN verification research in the ML community. There is no indication that the authors of many verification papers are aware of the FP unsoundness issues or strive to push for FP soundness in any way. Many verification papers (including both complete and incomplete ones) claim to provide some verification results, where the use of the word "verification" suggests the readers that their results are guaranteed to be correct, whereas in fact they provide no guarantee of correctness but totally skip the discussion of related FP soundness issues. Please find more details in our [comment below](https://openreview.net/forum?id=bVzUDC_4ls&noteId=bKAWgRoLfq) in response to Reviewer #1.
> > >
> > > The contrast between this "folklore in verification" and the reality of NN verification research where FP soundness issues have not received wide acknowledgement highlights the significance of our paper.

---

### Author Response · Authors · 2020-11-14
**Comparing a concurrent submission**

We thank all the reviewers for providing the insightful comments that help improve the quality of this work.

One concern appears to be the importance/relevance of the result presented in the paper (that the claims of neural network verifiers can be practically invalidated by exploiting floating point error).

We note that there is a concurrent ICLR submission also involving floating point issues in the verification of deep neural networks ([link](https://openreview.net/forum?id=4IwieFS44l)). The presence of the other submission, along with the fact that none of the reviewers of the other submission questions the importance or relevance of the topic, highlights the interest in this topic within at least part of the ICLR audience.

Having said that, there are important differences between these two papers:

* In this paper, we work with naturally trained networks to attack a complete verifier and manipulate network weights to attack an incomplete verifier. We explicitly recommend that sound verifiers should accurately (or conservatively) model the effects of any FP computations in the network inference or verification system. We also acknowledge incomplete verifiers that are sound with respect to FP computations in certain NN implementations. Our attack method for complete verifiers does not need to modify the network and does not assume any properties of the verifier other than that it unsoundly model FP computations while attempting to be complete. Furthermore, we present empirical analysis of the error behavior of different NN implementations. We also propose to quantize the network to obtain sound and complete verification that can be practically implemented, which is not discussed in the other paper.
* In the other paper, the authors modify the network weights to insert a backdoor that causes the network to give a different prediction when certain patterns are present in the input. The backdoor bypasses a complete verifier due to floating point error. They also propose an empirical defense by perturbing the network weights. Although the defence makes it more challenging (but not necessarily impossible) to insert backdoors, it does not solve the FP soundness issues in verifiers and the verification results can still be invalidated using our method. On the contrary, we advocate incorporating FP soundness into future verifiers to make the system truly trustworthy. Compared to our paper, the other paper does not discuss incomplete verifiers and does not analyze different FP characteristics of different NN implementations.

---

### Decision · Program_Chairs · 2021-01-07
**Final Decision**

**Decision:**

Reject

**Comment:**

There are many recent methods for the formal verification of neural networks. However, most of these methods do not soundly model the floating-point representation of real numbers. This paper shows that this unsoundness can be exploited to construct adversarial examples for supposedly verified networks. The takeaway is that future approaches to neural network verification should take into account floating-point semantics.

This was a borderline paper. On the other hand, to anyone well-versed in formal methods, it is not surprising that unsound verification leaves the door open for exploits. Also, there is prior work (Singh et al., NeurIPS 2018) on verification of neural networks that explicitly aims for soundness w.r.t. floating-point arithmetic. On the other hand, it is true that many adversarial learning researchers do not appreciate the value of this kind of soundness. In the end, the decision came down to the significance of the result. Here I have to side with Reviewer 1: the impact of this problem is limited in the first place, and also, the issue of floating-point soundness has come up in prior work on neural network verification. For these reasons, the paper cannot be accepted this time around.